Anthropogenic fertilization influences a shift in barley rhizosphere microbial communities

Enagbonma Ben Jesuorsemwen
Fadiji Ayomide Emmanuel
http://orcid.org/0000-0003-4344-1909 Babalola Olubukola Oluranti olubukola.babalola@nwu.ac.za
Food Security and Safety Focus Area, Faculty of Natural and Agricultural Sciences, North-West University , Mmabatho, North-West Province , South Africa
Okpala Charles
Electronic publication date: 2024 Jul 10
Publication date: 2024
Volume: 12
Electronic Location ID: e17303
Received 2023 Dec 11; Accepted 2024 Apr 4
Copyright: © 2024 Enagbonma et al.
Copyright year: 2024
Copyright holder: Enagbonma et al.
License: This is an open access article distributed under the terms of the Creative Commons Attribution License, which permits unrestricted use, distribution, reproduction and adaptation in any medium and for any purpose provided that it is properly attributed. For attribution, the original author(s), title, publication source (PeerJ) and either DOI or URL of the article must be cited.
License URL: https://creativecommons.org/licenses/by/4.0/

Keywords: Amplicon sequencing, Crop production, Metagenomics, Root exudate, Synthetic fertilizers

Funding: The National Research Foundation, South Africa UID135449; OOB The National Research Foundation, South Africa, funded this research (grant numbers UID135449; OOB). The funders had no role in study design, data collection and analysis, decision to publish, or preparation of the manuscript.

==============================
Background

Anthropogenic mediations contribute a significant role in stimulating positive reactions in soil–plant interactions; however, methodical reports on how anthropogenic activities impact soil microorganism-induced properties and soil health are still inadequate. In this study, we evaluated the influence of anthropogenic fertilization of farmland soil on barley rhizosphere microbial community structure and diversity, and the significant impacts on agro-ecosystem productivity. This will help validate the premise that soil amendment with prolonged synthetic fertilizers can lead to a significant reduction in bacterial abundance and diversity, while soils amended with organic fertilizers elicit the succession of the native soil microbial community and favor the growth of copiotrophic bacteria.

Methods

The total metagenomic DNA was extracted from soils obtained from the barley rhizosphere under chemical fertilization (CB), organic fertilization (OB), and bulk soil (NB). Subsequently, these samples were sequenced using an amplicon-based sequencing approach, and the raw sequence dataset was examined using a metagenomic rast server (MG-RAST).

Results

Our findings showed that all environments (CB, OB, and NB) shared numerous soil bacterial phyla but with different compositions. However, Bacteroidetes, Proteobacteria, and Actinobacteria predominated in the barley rhizosphere under chemical fertilization, organic fertilization, and bulk soils, respectively. Alpha and beta diversity analysis showed that the diversity of bacteria under organic barley rhizosphere was significantly higher and more evenly distributed than bacteria under chemical fertilization and bulk soil.

Conclusion

Understanding the impact of conventional and organic fertilizers on the structure, composition, and diversity of the rhizosphere microbiome will assist in soil engineering to enhance microbial diversity in the agroecosystem.

Introduction

Globally, crop production is presently increasing because of the high demand for animal feed, biological fuels and food. The increase in oil prices has made bioenergy more economical and cost-effective compared to fossil fuels. At present, 47.9 million km2 of land is dedicated to agriculture; this will certainly increase with the high population rate. The ever-increasing human population and the need to boost the production of cash and food crops have led to the anthropogenic amendment of soil with chemical fertilizers (Amoo et al., 2021; Hemathilake & Gunathilake, 2022). Chemical fertilizers encompass a high dose of major nutrients like phosphorous, potassium, and nitrogen, inorganic salts, and microelements like sulfur, magnesium, and calcium. The nutrient value in chemical fertilizers is specified as the N:P:K rate, signifying the proportions of nitrogen, total phosphorus, and potassium. These synthetic fertilizer inputs could impact farmland ecology and have additional influences on microbial variability (Enebe & Babalola, 2020). Yang et al. (2021) reported that long-term N fertilization greatly reduced microbial biomass, which can further influence the function and structure of the soil microbial community. It was also reported by Wakelin et al. (2012) that phosphorus application reliably shifted the structure and functionality of phosphorus cycling-associated soil microorganisms, as the insufficiency of accessible soil phosphorus elicits the bacterial community to develop increased phosphorus solubilization ability. It has been reviewed that too much and perpetual application of chemical fertilizers vastly impacts soil nutrients (Pahalvi et al., 2021), quickens soil acidification (Hao et al., 2020), intensifies the decline of soil fertility (Bhatt, Labanya & Joshi, 2019), influences soil microbial communities (Luan et al., 2020) and leads to environmental deterioration (Kumar, Kumar & Prakash, 2019).

Due to the decline in microbial population and environmental problems (such as greenhouse gas emissions, pollution of water, and soil ecosystems) linked with chemical fertilizer, as well as the quest to realize optimum farming without compromising future generations, agronomists are assessing the need to use organic fertilizers to boost crop production (Trujillo-Tapia & Ramírez-Fuentes, 2016). Organic fertilizers are derived from animal or plant-based resources or other organic elements that are the product of naturally occurring processes. These fertilizers contain both major and minor nutrients essential for plant growth (Tan et al., 2023). They can also be an active source of soil microbes while also improving soil structure (Sharma, Datta & Sharma, 2023). Lin et al. (2019) reported that applying organic fertilizer can shape the microbial composition and recruit useful bacteria into the tea rhizosphere. This became evident from their results, which showed a significant increase in the relative abundance of Gemmatimonadales, Acidobacteriales, Solibacterales, Streptomycetales, Burkholderiales, Nitrospirales, Ktedonobacterales and Myxococcales. The investigation of Underwood et al. (2011) revealed that the variety of fungi, archaea, viruses and bacteria is increased in soil due to the rich nutrients in organic manure added to the soil. This claim became clear from rhizosphere microbiome research conducted by Qiao et al. (2019), who demonstrated that organic fertilization promotes greater bacterial diversity than chemical fertilization.

The rhizospheres of plants have evolved into diverse and complex microbial groups with different information processing systems involved in plant enlargement and growth and plant defence response (Babalola et al., 2020). Plants induce substantial selection pressure on the development of some bacteria, such as Rhizobium, accomplished through the emission of roots’ exudates (Enagbonma et al., 2023). The secreted root exudates contain various compounds that attract the development of specific plant microbiota. The attracted organisms utilize these exudates as sources of energy and multiply in their vicinity (Bukhat et al., 2020; Pantigoso, Newberger & Vivanco, 2022).

Globally, barley (Hordeum vulgare) is cultivated on about forty-eight million acres of land and is the fourth most grown grain. Furthermore, it is an excellent investigational model for studying plant-microbe communications in the light of domestication and crop selection. (Escudero-Martinez et al., 2022; Giraldo et al., 2019). Barley plays a significant role in the selection, fortification, and nourishment of rhizosphere microbial composition and structure (Verstegen et al., 2014). The barley rhizosphere microbiome plays a significant function in improving plant fitness and is crucial for disease suppression and participating in soil biogeochemical cycling (Berendsen, Pieterse & Bakker, 2012; Lu et al., 2018). It is also a primary caveat (pointer) of soil quality due to its quick reaction to ecological alteration (Zheng et al., 2020).

The bacterial composition, structure, and diversity are posited to underpin ecosystem functioning, and their loss can adversely impact soil health and food security (Bano, Wu & Zhang, 2021). To comprehend the adverse, neutral, or positive effects of fertilization schemes on soil microbiomes, we use the amplicon sequencing approach to profile the bacterial structure, richness, and variability of barley rhizosphere. The method has been used to provide complete insights into the species diversity of microbial communities in soil systems (Amoo & Babalola, 2019). This study hypothesis assumes that soil amendments with chemical fertilizers are often lethal to non-target soil microorganisms, and organic fertilizers elicit succession of the native soil microbial community. We also posited that organic fertilization would favor the growth of copiotrophic bacteria, while long-term chemical fertilization causes a significant reduction in bacterial abundance and biodiversity.

Materials and Methods

Study area and sample collection

Six weeks after the germination of the barley seed, we collected twenty-four soil samples of 20 g each from barley rhizosphere under chemical fertilization (eight soil samples) (25°39″32.2″S 27°39′49.8″E), barley rhizosphere under organic fertilization (eight soil samples) (25°39′04.9″S, 27°40′46.6″E), and bulk soils which served as the experimental controls (eight soil samples), at a depth of 0–5 cm. Data were collected as previously described in Babalola & Enagbonma (2024). The history of the soil’s physical and chemical properties (Table S1) before fertilization and planting was done following the standard analytical method earlier reported by Enagbonma, Amoo & Babalola, 2021.

Molecular and downstream analysis

The method used for the DNA extraction, library preparation of the16S rRNA gene, generated reads after quality control (QC), and the taxonomic analysis of the soil samples under different fertilization schemes (CB = 8 samples, OB = 8 samples) and the bulk soils (NB = 8 samples) were done as previously described in Babalola & Enagbonma (2024). The metagenomic rast server technology presented an approximation of microbial abundances existing in the barley rhizosphere under chemical fertilization (CB), barley rhizosphere under organic fertilization (OB), and bulk soil (NB). After the MG-RAST analysis with default settings (Jiang & Takacs-Vesbach, 2017; Keegan, Glass & Meyer, 2016) on the 24 individual sequences, we computed the comparative abundances of the taxa via percentages. Thereafter, the mean number of relative abundances of the eight replicates for each site (CB, OB, and NB) was used for statistics.

From this statistical analysis, an assessment was performed to evaluate the composition, structure, and diversity of the microbiome among the samples. The normalization tool in metagenomic rast server technology was switched on to standardize the dataset. The rarefaction curve analysis was also prepared via the rarefaction tool in MG-RAST. PAST version 3.20 (Hammer & Harper, 2001) was used to evaluate alpha diversity (via Pielou Evenness, Simpson, Chao-1, and Shannon) for individual samples. The Kruskal–Wallis test was used to compare these indices among sites. The principal coordinate analysis (PCoA) was used to depict the diversity between species (beta diversity) based on a Euclidean distance matrix, and the differences in community composition among the groups of samples were tested by using the one-way analysis of similarity through 999 permutations (Clarke, Somerfield & Chapman, 2006). The principal component analysis (PCA) demonstrated how the bacteria at the phylum level were distributed among sites. The PCoA and PCA were designed by CANOCO 5 (Šmilauer & Lepš, 2014). The Shinyheatmap was used to plot the heatmap with a z-score converted to the relative abundance of bacterial classification (Khomtchouk, Hennessy & Wahlestedt, 2017).

Results

Sequencing information and bacterial distribution across the barley rhizosphere under different fertilization schemes and bulk soils

The sequence information of the 24 samples for barley rhizosphere under chemical fertilization (CB), organic fertilization (OB), and Bulk soil (NB) are summarized in Table S3.

Analysis of the amplicon sequence data using the RDP database showed that 22 phyla were present in barley rhizosphere obtained from CB and OB as well as the bulk soil (NB), and the others were grouped as unclassified bacteria (Fig. 1). PCA was plotted to demonstrate how these bacteria at phylum level were distributed among the barley rhizosphere (CB and OB) and the bulk soil with PCA axes 1 and 2 elucidated 95.50% and 4.50% variance correspondingly (Fig. 2). The arm length of the PCA revealed that Bacteroidetes, Verrucomicrobia, Nitrospirae, Planctomycetes, Spirochaetes, Aquificae, Dictyoglomi, Fibrobacteres, and Synergistetes predominated in the barley rhizosphere under chemical fertilization (CB) while Proteobacteria, Firmicutes, Cyanobacteria, Fusobacteria, Tenericutes, Deinococcus-Thermus and Chlorobi predominated in the barley rhizosphere under organic fertilization (OB). The bulk soil contained predominantly with Actinobacteria, Acidobacteria, Chloroflexi, Chlamydiae, Thermotogae, and Thermodesulfobacteria. The relative abundance (phylum level) of Proteobacteria (18.95%) in barley rhizosphere under organic fertilization (OB) was significantly higher than the Proteobacteria (18.58%) of barley rhizosphere under chemical fertilization (CB) and the bulk soil (NB) (17.52%). The relative abundance of Actinobacteria were significantly higher in NB (22.42%) than the relative abundances of Actinobacteria in CB (19.14%) and OB (18.41%) while the relative abundance of Bacteroidetes was significantly higher in CB (4.17%) than the relative abundances of OB (3.87%) and NB (2.71%) (Fig. 1). At the species level, 2,320 species were recorded in the barley rhizosphere under chemical fertilization (CB), while 2,393 species were recorded in the barley rhizosphere under organic fertilization (OB) and 2,197 species were observed in bulk soil (NB). Rubrobacter radiotolerans dominated in CB, while Bacillus megaterium dominated in OB and Rubrobacter xylanophilus dominated the bulk soil NB (Fig. 3).

Figure 1 Bacterial phylum (relative abundance) in barley rhizosphere under chemical and organic fertilization regimes and bulk soil.

The scale block denotes the colour concentration gradient that represents the relative abundance of the bacterial taxa as converted by the Z-score. CB = barley rhizosphere under chemical fertilization, OB = barley rhizosphere under organic fertilization, NB = bulk soil.

Figure 2 Principal component analysis (PCA) of bacterial distributions.

The arm length symbolizes the strength of distribution of the bacteria. The PCA axes 1 and 2 elucidated 95.50% and 4.50% variation, correspondingly. CB = barley rhizosphere under chemical fertilization, OB = barley rhizosphere under organic fertilization, NB = bulk soil.

Figure 3 Bacterial species (relative abundance) in barley rhizosphere under chemical and organic fertilization regimes and bulk soil.

The scale block denotes the colour concentration gradient that represents the relative abundance of the bacterial taxa as converted by the Z-score. CB = barley rhizosphere under chemical fertilization, OB = barley rhizosphere under organic fertilization, NB = bulk soil.

Assessment of bacterial diversity from the barley rhizosphere and the bulk soil

The alpha diversity for the barley rhizosphere under fertilization and bulk soil was calculated (Chao-1) to be 2,320 species in CB, 2,393 species in OB, and 2,197 species in NB. The Simpson, Shannon index and Evenness values were significantly higher in OB, followed by CB and NB (Table 1). The rarefaction curve (Fig. S1) shows that the bacterial richness was higher in OB when compared with CB and NB. Contrasting among any duo of bacterial societies (beta diversity) using principal coordinate analysis (Fig. 4) revealed no clustering in the compared environments. The analysis of similarity calculated the p-value to be 0.02 and the R-value to be 0.67, suggesting that the separation of sites is strong. For example, sample OB is separate and far away from sample NB, meaning that its bacterial society and structure are distinctive from those of the bulk soils. Sample OB was far away from CB, signifying that the bacterial society and structure between the two samples differ (Fig. 4).

Table 1 Estimation of alpha diversity of barley rhizosphere and bulk soil samples.

Diversity indices	CB	OB	NB	p-value	
Simpson_1-D	0.9791	0.98	0.9758		
Shannon_H	4.941	5.009	4.816	0.006	
Evenness_e^H/S	0.06031	0.06249	0.05621		
Chao-1	2,320	2,396	2,197		
Note:

CB = barley rhizosphere under chemical fertilization, OB = barley rhizosphere under organic fertilization, NB = bulk soil.

Figure 4 PCoA plot of bacterial composition of barley rhizosphere and bulk soil.

CB = barley rhizosphere under chemical fertilization, OB = barley rhizosphere under organic fertilization, NB = bulk soil.

Discussion

This study used the amplicon sequencing technique to profile the bacterial composition, abundance, and diversity of barley rhizosphere under different fertilization regimes (CB and OB) and to see if there was a marked shift from comparable bulk soils. Soil fertilization is an old farming scheme targeted at promoting the fertility of soils for high growth and yield of crops (Bitire et al., 2022; Seenivasagan & Babalola, 2021). Lately, ecologists have concentrated on revealing the influence of fertilisation regimes on the soil’s microbial societies (Ajilogba et al., 2022; Masowa et al., 2021). This investigation was able to establish that some substantial differences occur in bacteria from barley rhizosphere under chemical fertilization, organic fertilization, and bulk soil.

PCA (Fig. 2) supported our assumption that each site (CB, OB, and NB) has a predominant bacteria phylum. This bacterial variation could be linked to the fertilization scheme which adds nutrients to the soil but alters the soil pH (Ajilogba, Habig & Babalola, 2022; Zhang et al., 2017). This variation was also seen in bacterial composition with the relative abundance of Bacteroidetes, Proteobacteria and Actinobacteria predominating CB, OB and NB respectively (Fig. 1). This finding supported the work of Zhang et al. (2022) who reported that the application of mixed organic and inorganic fertilizers drives bacterial community changes in teak plantations. Although Proteobacteria (copiotrophic phyla) dominated the barley rhizosphere under organic fertilization (OB), our study does not provide sufficient evidence to confidently state that soils amended with organic fertilizers exclusively favor the growth of copiotrophic bacteria. This is due to the dominance of another copiotrophic phylum, Bacteroidetes, in the barley rhizosphere under chemical fertilization (CB). Our finding was supported by Luo et al. (2023) who reported that both organic and mineral nutrient inputs promoted copiotroph-dominated bacterial assemblages (including Proteobacteria and Bacteroidetes members). The alterations in bacterial dominance among the sites may perhaps impact the ecosystem functions contributed by these bacteria (Enagbonma, Aremu & Babalola, 2019; Enebe & Babalola, 2022). It also indicated a clear shift in the bacterial community structure in the rhizosphere under different fertilization practices, however it is not clear whether if this temporal selection is triggered by the barley plants’ selective pressure or the bacteria.

Rhizosphere soil under organic fertilization showed the highest alpha diversity (evenness and richness) of bacteria (Table 1, Fig. S1). This supported a previous study that reported that organic fertilization commonly increases bacterial abundance in soils with manure (Cheng et al., 2020; Uzoh et al., 2021). This was also reflected in the total number of species recorded in each site, with 2,393 species found in OB, 2,320 species found in CB, and 2,197 species in bulk soil (Fig. 3). This pattern was also observed by Enebe & Babalola (2020) when they used shotgun metagenomics to profile the bacterial richness and structure in maize rhizosphere under different fertilization schemes.

These bacterial variations among CB, OB, and NB were further supported by the beta diversity analysis via PCoA (Fig. 4), suggesting that the separation of sites is significantly strong (p < 0.05). For example, sample OB was separate and far away from sample NB, meaning that its bacterial community and structure are unique from those of the bulk soils. Sample OB was away from CB, signifying that its bacterial society and structure between the two samples differ (Fig. 4). Our rarefaction curve (Fig. S1) shows that the metagenome of the organic fertilizer-treated rhizosphere soil and the control samples were alike. This suggests that organic fertilizer showed a more stable community than the chemical fertilizer. However, we are aware of the barley plant selection effect, which could be credited to chemical signalling compounds secreted by the plants’ roots through rhizodeposition (Bouhaouel et al., 2019; Liljeroth et al., 1990). The correlation between the microbial community enrichment effects of organic manure and barley plants are both capable of promoting the soil organic carbon needed by bacteria.

We expected the bacterial diversity in CB to be lower than in NB, but this was not the case, as revealed by the alpha diversity analysis (Table 1). Therefore, care must be taken in relating amended soils that are inorganic vs organic. In all, this study revealed that effects on the abundance, structure, and diversity of the rhizosphere microbiomes are governed by fertilization. Fertilization supervision is recommended to manipulate rhizosphere bacterial communities to farmers’ advantage.

Conclusions

Taken together, this research unveiled and supported the various studies that stated that soil fertilization brings about a microbial shift. These bacterial differences were also supported by the alpha and beta diversity analysis, which showed that bacteria under organic fertilization were more diverse and evenly distributed when related to the bacteria under chemical fertilization and bulk soils. Moreover, this study revealed the dominance of Bacteroidetes, Proteobacteria, and Actinobacteria in barley rhizosphere under chemical fertilization, organic fertilization, and bulk soils, respectively, suggesting the shift in their ecological function played by these bacteria. Results from this study, coupled with the imperative to feed the ever-increasing human population, underscore the need to adopt sustainable integrated fertilizer strategies. These approaches are crucial for optimizing outputs, maintaining microbial communities, and promoting plant health and yield.

Supplemental Information

Supplemental Information 1 Supplemental Figure and Tables.

The authors thank North-West University and members of the Food Security and Safety Focus Area for the conducive environment afforded them during the research.

Additional Information and Declarations

Competing Interests

Author Contributions

DNA Deposition

Data Availability

The authors declare that they have no competing interests.

Ben Jesuorsemwen Enagbonma conceived and designed the experiments, performed the experiments, analyzed the data, prepared figures and/or tables, authored or reviewed drafts of the article, and approved the final draft.

Ayomide Emmanuel Fadiji conceived and designed the experiments, performed the experiments, analyzed the data, prepared figures and/or tables, authored or reviewed drafts of the article, and approved the final draft.

Olubukola Oluranti Babalola conceived and designed the experiments, performed the experiments, analyzed the data, prepared figures and/or tables, authored or reviewed drafts of the article, and approved the final draft. Babalola OO supervised the other two authors.

The following information was supplied regarding the deposition of DNA sequences:

The sequences used for this study are available in the SRA of the NCBI: PRJNA827679 (CB1), PRJNA827686 (CB2), PRJNA827693 (CB3), PRJNA827699 (CN4), PRJNA827706 (CB5), PRJNA827761 (CB6), PRJNA827780 (CB7), PRJNA827786 (CB8), PRJNA826806 (OB1), PRJNA826824 (OB2), PRJNA826834 (OB3), PRJNA826841 (OB4), PRJNA826853 (OB5), PRJNA827254 (OB6), PRJNA827256 (OB7), PRJNA827257 (OB8), PRJNA828106 (NB1), PRJNA828099 (NB2), PRJNA828045 (NB3), PRJNA828037 (NB4), PRJNA828022 (NB5), PRJNA828017 (NB6), PRJNA828010 (NB7), and PRJNA828003 (NB1) for barley rhizosphere samples under chemical fertilization (CB), organic fertilization (OB) and for the bulk soil (NB) samples.

The following information was supplied regarding data availability:

The sequences used for this study are available in the SRA of the NCBI: PRJNA827679 (CB1), PRJNA827686 (CB2), PRJNA827693 (CB3), PRJNA827699 (CN4), PRJNA827706 (CB5), PRJNA827761 (CB6), PRJNA827780 (CB7), PRJNA827786 (CB8), PRJNA826806 (OB1), PRJNA826824 (OB2), PRJNA826834 (OB3), PRJNA826841 (OB4), PRJNA826853 (OB5), PRJNA827254 (OB6), PRJNA827256 (OB7), PRJNA827257 (OB8), PRJNA828106 (NB1), PRJNA828099 (NB2), PRJNA828045 (NB3), PRJNA828037 (NB4), PRJNA828022 (NB5), PRJNA828017 (NB6), PRJNA828010 (NB7), and PRJNA828003 (NB1) for barley rhizosphere samples under chemical fertilization (CB), organic fertilization (OB) and for the bulk soil (NB) samples.

https://www.ncbi.nlm.nih.gov/bioproject/?term=PRJNA827679 (CB1), https://www.ncbi.nlm.nih.gov/bioproject/?term=PRJNA827686 (CB2), https://www.ncbi.nlm.nih.gov/bioproject/?term=PRJNA827693 (CB3), https://www.ncbi.nlm.nih.gov/bioproject/?term=PRJNA827699 (CB4), https://www.ncbi.nlm.nih.gov/bioproject/?term=PRJNA827706 (CB5), https://www.ncbi.nlm.nih.gov/bioproject/?term=PRJNA827761 (CB6), https://www.ncbi.nlm.nih.gov/bioproject/?term=PRJNA827780 (CB7), https://www.ncbi.nlm.nih.gov/bioproject/?term=PRJNA827786 (CB8), https://www.ncbi.nlm.nih.gov/bioproject/?term=PRJNA826806 (OB1), https://www.ncbi.nlm.nih.gov/bioproject/?term=PRJNA826824 (OB2), https://www.ncbi.nlm.nih.gov/bioproject/?term=PRJNA826834 (OB3), https://www.ncbi.nlm.nih.gov/bioproject/?term=PRJNA826841 (OB4), https://www.ncbi.nlm.nih.gov/bioproject/?term=PRJNA826853 (OB5), https://www.ncbi.nlm.nih.gov/bioproject/?term=PRJNA827254 (OB6), https://www.ncbi.nlm.nih.gov/bioproject/?term=PRJNA827256 (OB7), and https://www.ncbi.nlm.nih.gov/bioproject/?term=PRJNA827257 (OB7)

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
