# Peer review of "Anthropogenic fertilization influences a shift in barley rhizosphere microbial communities"

_PeerJ, doi:10.7717/peerj.17303_

## Round 0.1 · original submission · Major Revisions

Thank you authors for your patience whilst your work was examined by the reviewers, who have found a lot of merit and provided useful suggestions.

Please address the comments, especially the annotated manuscripts pdf attached, which the reviewers have kindly provided.

Make a diligent effort to address all concerns, please ensure to provide detailed responses in the reply to comments, as well as in the revised manuscript. Corrections in these areas are essential to enhance the overall quality and suitability of the submission for publication.

**Language Note:** PeerJ staff have identified that the English language needs to be improved. When you prepare your next revision, please either (i) have a colleague who is proficient in English and familiar with the subject matter review your manuscript, or (ii) contact a professional editing service to review your manuscript. PeerJ can provide language editing services - you can contact us at [email protected] for pricing (be sure to provide your manuscript number and title). – PeerJ Staff

·

Basic reporting

Pay attention to the comments provided by the reviewer in the attached file and make corrections based on them

Experimental design

Pay attention to the comments provided by the reviewer in the attached file and make corrections based on them

Validity of the findings

Pay attention to the comments provided by the reviewer in the attached PDF file and make corrections based on them

Additional comments

Pay attention to the comments provided by the reviewer in the attached PDF file and make corrections based on them

Reviewer 2 ·

Basic reporting

General comments:
This manuscript focuses on investigating the changes in barley rhizosphere microbial communities by human fertilization influences. This research is original, and the experimental design is good. The statistical methods employed are robust and sufficient to address the research questions.

Specific Comments (Please see the line number annotated document):
Abstract:
I recommend the authors check the Abstract requirements with the editor. Generally, the Abstract should be one paragraph with a word limit.
Introduction:
The background information connects well with the study. Research questions are well defined and relevant.
L72: There is extra space in this line.
L85: “barley”. It is the first time the authors mentioned “barley”. As a scientific paper, the authors should give barley’s scientific name.

Materials & Methods:
Generally, more information is needed to clarify the experiments and molecular methods.
L106-121: In this paragraph, the authors should add a sentence and explain the purpose of sampling “bulk soil”. It is not clear whether the bulk soil samples are experimental controls or not.
L123-124: “We used the Nucleospin Soil kit (Macherey-Negel, Germany) to extract the DNA of the entire barley rhizosphere microbiome by using the kit’s manual as a guide”.
It is unclear the number of samples extracted from the DNA. The authors mentioned a total of 24 samples collected in L106-121. Were all these samples used for the DNA extraction?
L127: “universal primers”. Please include the citation of these primers.
L130: “MiSeq was used to sequence the quantified libraries”. Some information is missing here. What kind of MiSeq sequencing was used? Was it 2X300 bp or 2X250 bp? Also, the authors should include where the sequencing was carried out (the name of the sequencing center).
L132: “MG-RAST server”. If the authors used MG-RAST, a list of MG-RAST sample access numbers/IDs should be given somewhere in the manuscript.
Results:
L163: “Table S2”. Please see the comments above (the number of samples for the molecular analysis). Table S2 only shows the three different types of soil. It is not clear how many samples were sequenced for each type of soil.

Discussion:
In general, an in-depth discussion is needed. For example, the authors hypothesized that “organic fertilization will favor the growth of copiotrophic bacteria … (L102)”. Do current results support this hypothesis? Are there any copiotrophic bacteria in the organic fertilization treatment? Are current results consistent with the results from previous soil microbiome studies?
L229-230: The font in this sentence is not consistent with the rest of the manuscript.

Experimental design

NA

Validity of the findings

NA

Additional comments

NA

Annotated reviews are not available for download in order to protect the identity of reviewers who chose to remain anonymous.

---

## Round 0.2 · Minor Revisions

Authors, please kindly attend to the very minor observations before the work can be accepted for publication. A reviewer has identified them (annotated) to guide you through. Thank you

·

Basic reporting

It is well done

Experimental design

It is well done

Validity of the findings

It is well done

Additional comments

It is well done

Reviewer 2 ·

Basic reporting

Dear Editor,
The manuscript has been significantly improved after the revision. The current manuscript’s writing is clear, and the research is novel. The authors have addressed all the questions from the first review. Therefore, I recommend the editor to accept this manuscript.
I also recommend authors should include some important bioinformatic software citations and parameters before officially publishing this manuscript. Please see the details below.

L119: “After the MG-RAST…”. Please cite the MG-RAST paper below
Keegan KP, Glass EM, Meyer F (2016) MG-RAST, a Metagenomics Service for Analysis of Microbial Community Structure and Function. Methods Mol Biol 1399:207-233
On the MG-RAST platform, if the authors used MG-RAST with “default settings”, the authors should mention it here. Apart from the e-value, default settings include a minimum identity and a minimum alignment length. These are key parameters. Please consider citing and checking the paper below and see how to report these key parameters.
Jiang, X., and Takacs-Vesbach, C.D. (2017) Microbial community analysis of pH 4 thermal springs in Yellowstone National Park. Extremophiles 21: 135-152.
L125: “PAST version 3.20 was used to evaluate alpha ….”
Please consider citing the software PAST paper below
Hammer Ø, Harper DA (2001) Past: paleontological statistics software package for education and data analysis. Palaeontologia electronica 4:1
L 133: “CANOCO 5 (http://www.canoco5.com/).”
Please consider replacing the website with the official publication below
Smilauer P, Leps J (2014) Multivariate analysis of ecological data using CANOCO 5. Cambridge university press

Experimental design

NA

Validity of the findings

NA

Additional comments

NA

Annotated reviews are not available for download in order to protect the identity of reviewers who chose to remain anonymous.

---

## Round 0.3 · accepted · Accept

Thank you authors for revising your work. I am very satisfied with the current revised version. It is acceptable for publication. Very much appreciate the authors for finding PeerJ as your journal of choice and look forward to your future scholarly contributions. Congratulations.